# Hardware Design and Accurate Simulation of Structured-Light Scanning for Benchmarking of 3D Reconstruction Algorithms

**Sebastian Koch**\*
TU Berlin
s.koch@tu-berlin.de

**Yurii Piadyk**\*
New York University
ypiadyk@nyu.edu

**Markus Worchel**
TU Berlin
m.worchel@campus.tu-berlin.de

**Marc Alexa**
TU Berlin
marc.alexa@tu-berlin.de

**Claudio Silva**
New York University
csilva@nyu.edu

**Denis Zorin**
New York University
dzorin@cs.nyu.edu

**Daniele Panozzo**
New York University
panozzo@nyu.edu

## Abstract

Images of a real scene taken with a camera commonly differ from synthetic images of a virtual replica of the same scene, despite advances in light transport simulation and calibration. By explicitly co-developing the Structured-Light Scanning (SLS) hardware and rendering pipeline we are able to achieve negligible per-pixel difference between the real image and the synthesized image on geometrically complex calibration objects with known material properties. This approach provides an ideal test-bed for developing and evaluating data-driven algorithms in the area of 3D reconstruction, as the synthetic data is indistinguishable from real data and can be generated at large scale by simulation. We propose three benchmark challenges using a combination of acquired and synthetic data generated with our system: (1) a denoising benchmark tailored to structured-light scanning, (2) a shape completion benchmark to fill in missing data, and (3) a benchmark for surface reconstruction from dense point clouds. Besides, we provide a large collection of high-resolution scans that allow to use our system and benchmarks without reproduction of the hardware setup on our website:

https://geometryprocessing.github.io/scanner-sim

## 1 Introduction

Reconstruction of 3D geometry from real world measurements is a fundamental task in computer vision and data-driven methods are emerging as a promising approach to increase the reliability and quality of the reconstruction by leveraging data priors to compensate for noise or missing data. Due to the wide availability of low cost and low accuracy scanners, most of the benchmarks, datasets and algorithm development in 3D reconstruction rely on this type of data. In this work, we propose the first large scale data generator, dataset, and benchmarks for high resolution and high accuracy 3D reconstruction from structured-light scanning.

---

\*Both authors contributed equally to this work.

35th Conference on Neural Information Processing Systems (NeurIPS 2021) Track on Datasets and Benchmarks.

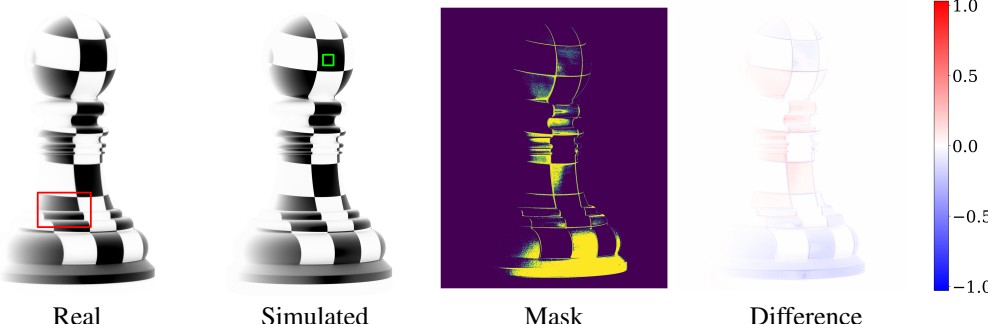

| Real | Simulated | Mask | Difference |

Figure 1: From left to right: the image of the *Pawn* (inverted colors) acquired by a camera is faithfully reproduced by our scanning simulator after hardware calibration. The mask shows that pixel-wise difference rarely exceeds $5\%$ of the average brightness in the middle of the white checker highlighted by the green rectangle which is used to normalize the images (i.e. $100\%$, or $1.0$). More information about remaining sources of errors can be found in Section 7 with a close-up view of the highlighted red region shown in Figure 17. The height of this model is $152.5\,\mathrm{mm}$.

With rapid improvements in consumer sensors and 3D reconstruction methods the demand for high-quality 3D reconstruction will also grow rapidly. The high resolution regime poses a different set of challenges compared to low-accuracy 3D reconstruction, and its objective evaluation is more challenging, as it is impossible to rely on the data from a high resolution scanner as the ground truth, as done in some existing benchmarks (Section 2). As an alternative, we propose an approach in which, instead of attempting to acquire ground truth data at yet higher resolution, which is increasingly impractical, we use objects for which ground truth is known, as these are fabricated at high accuracy from CAD models. As the number of objects that can be precisely fabricated and scanned is necessarily limited, we couple this fabrication-based approach with an advanced simulator, calibrated using these high-accuracy real-world objects, capable of producing an unlimited amount of synthetic data closely approximating what our hardware setup would produce. We have three major contributions:

**1) Data Generator.** We introduce a hardware design for a structured-light 3d scanner paired with a simulator that produces accurate replicas of the images acquired by the scanner camera after calibration (Figure 1). This is achieved by combining careful hardware choices, modern light transport algorithms, precisely fabricated calibration geometry, and restricting the material and lighting setting. The outcome is a system that allows to create 3D scanning datasets for structured-light (a popular choice for both high-end photogrammetry scanners, and for the low cost real-time scanners commonly used in vision and robotics applications such as the Kinect or RealSense sensors) targeting both the high-resolution and high-accuracy regime, and low-accuracy settings, where the high accuracy of the simulator can then be used to quantitatively measure reconstruction errors and capture real-world sources of noise.

**2) Dataset of Real Scans.** We provide a dataset of real scans equipped with ground-truth geometry, which enable to use our data generator and benchmarks without requiring to reproduce the hardware setup. Any algorithm developed, trained, and evaluated with our system, can then be tested on this dataset of real scans to evaluate its generalization to real-data with minimal efforts. The dataset contains 4 precisely-machined calibration objects and 7 color-textured 3D printed objects scanned with 30 degrees rotating stage intervals, for a total of $\approx 0.5$ terabytes of data.

**3) Benchmark Tasks.** We combine our data generator and real scans to introduce three benchmarks for different surface reconstruction tasks of increasing complexity: a. filtering of scanner noise from a single range scan, b. completion of missing parts from a single range scan, and c. conversion of a collection of range scans into a triangulated surface. The first two tasks are ideal for convolutional neural networks that can exploit the regularity of the range scans and treat them as images, while the last task is much more challenging since it needs to support varying in- and output data sizes, and has both (a) and (b) as subtasks. For each benchmark we provide synthetic training and test data, real-world test data (to evaluate generalization), a procedure for evaluation, and the implementation of a baseline method.

The hardware blueprint for the scanner, the manufacturing protocol for the calibration objects, the reference implementation of the simulator, the datasets, and the benchmarks are available at https://geometryprocessing.github.io/scanner-sim.

## 2 Related Work

We refer to the additional material for an overview of structured light scanning hardware and reconstruction algorithms. We focus here specifically on datasets and benchmarks for 3D scanning.

**Stereo Reconstruction.** A popular dataset in stereo reconstruction has been introduced in [19] by constructing a stereo scanner and combining it with a projector to collect ground truth annotations using structured light. This approach however is challenging to scale to the generation of large datasets as the acquisition time and effort is high. This motivated the development of the SyB3R data generators by [13] which uses a photorealistic rendering system to synthesize annotated data. However, these synthetic datasets still can not replace real word datasets but complement them by guiding the design of the physical setup. A number of setups use additional mechanical components, such as robotic arms [9] or a spherical gauntry [20]; while substantially expanding the size of the dataset that can be obtained, these approaches might introduce additional sources of errors [4]. Our work aims to combine the best of both worlds, allowing acquisition of real data and the synthetic generation of data that is *indistinguishable* from the one acquired in the real world, opening the door to train models on synthetic data and to test them on real data. Extending our work to generate stereo data is one of the avenues of future work, as it will only require us to add an additional camera to our system. In fact, a variety of configurations can be tested without the need to build the actual error-prone physical replicas because of the high quality simulation of these setups.

**Laser and Structured Light.** [1] introduced a synthetic benchmark for surface reconstruction algorithms, using a simplified rendering model simulating a laser scanner. The approach is validated by 3d printing and scanning an object with a NextEngine laser scanner and qualitatively comparing the results. They do not *calibrate* their synthetic scanner to exactly match the NextEngine scanner, in part due to the lack of details on the internals of the NextEngine scanner. [14] present a similar synthetic benchmark for structured light reconstruction algorithms with the goal of measuring the effects of illumination artifacts, including projector defocus, inter-reflections and subsurface scattering. They build a synthetic simulator using photo-realistic rendering (implemented with PBRT) that takes an object and its BRDF into account. They show that some of the discoveries made with the simulator apply to a physical scanner, but there is no attempt to match the results of the simulator and of the real scanner. To the best of our knowledge, our work is the first proposing a simulator that creates indistinguishable results from the physical scanner, thus allowing to generate large, faithful training datasets without having to scan manually thousands of objects.

**Synthesis of Realistic Images.** In terms of side-by-side comparisons between rendered and photographed images, Phong's [17] seminal paper was first in using visual comparison of the rendered image of a sphere to a photograph to highlight the quality of his shading model. Meyer et al. [15] performs two detailed studies: comparing radiometric measurements between physical and rendered models, and a perceptual study comparing rendered images shown on a color TV monitor to the physical model using the Cornell Box [5]. Pattanaik et al. [16] calibrates a CCD camera to compare real and synthetic imagery of the Cornell Box, and attribute image differences to "mismatch between the numerical description of the scene geometry and the actual geometry". We are not aware of any existing work able to achieve a faithfulness comparable to our approach, especially on geometrically complex objects. The problem of designing a perceptual model to compare real and synthetic images has been pioneered by Rushmeier et al. [18]. In our setting, we opt for direct pixel-wise difference as our goal is to generate replicas of images to faithfully simulate a 3D reconstruction instead of producing perceptually similar images.

# 3   Hardware Setup and Simulator

A structured-light scanning setup is composed of 3 main components: a camera $C$, a projector $P$, and the object being scanned $O$, which can be optionally placed on a rotating stage $S$ (Figure 2). We refer to our technical design document and to [12] for a more detailed introduction to structured-light scanning.

**SLS Primer.**   Assuming that the position of all the objects in the scene is known to sufficient accuracy, a 3D object can be reconstructed by illuminating a single pixel of the projector at a time, detecting the location of the illuminated point on the camera sensor and then triangulating. This procedure is impractically slow. By projecting a set of coded patterns it is possible to establish correspondences between camera and projector pixels from a small set of images [2, 6]. The accuracy of the reconstruction depends on many factors, including the resolution of the camera and projector, the lenses used, and the accuracy of the estimation of the relative position of camera and projector. Finding this set of parameters (calibration of the scanner) is usually performed by 'scanning' objects of known geometry. We carefully analyze each component of the scanning system, selecting hardware components to minimize the noise that we cannot simulate, and we propose a corresponding calibration procedure with the goal of minimizing reconstruction errors. Different from existing approaches, we do not strive to make the calibration procedure simple and/or efficient, our goal is solely on minimizing effects that cannot be recovered by optimization methods in the computational part of the system.

**Hardware.**   To build a highly accurate structured light scanner, we use the following parts: (1) A *CharuCo board* [3] for accurate calibration of the camera and projector geometry; (2) A *linear stage* capable of accurately reproducing positions for the CharuCo calibration board with known intervals to measure the focus distance and the aperture for both the

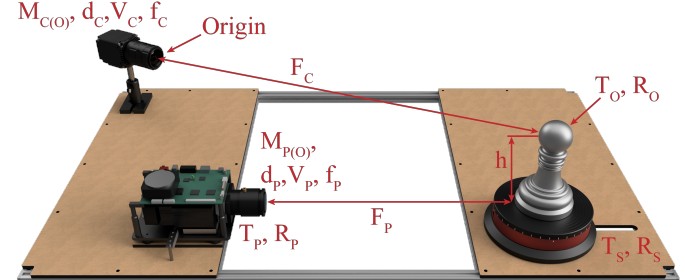

Figure 2: A diagram illustrating our structured light scanner setup and the parameters associated with each component (see supplemental material for a detailed explanation).

camera and the projector; (3) A *Spectralon®* for the radiometric calibration; (4) A set of precisely-machined calibration objects; (5) An Atlas 31.4 MP Camera by Lucid Vision Labs with Edmund Optics APS-C 50 mm lens and (6) a Texas Instruments DLP4710EVM-LC projector.

**Simulation.**   We base our rendering system on the physically based renderer MITSUBA [8], extending it to accurately model our specific motorized stages, camera, sensor, lenses, and to simulate our projector light source. Due to space limitations, we refer to our technical design document for a complete description of how each hardware component in our system is parametrized in Mitsuba and how each parameter is calibrated. In total there are 31 parameters, listed in Table 1 in the technical design document.

# 4   Quantitative Evaluation of Virtual and Physical Setups

We quantitatively evaluate the accuracy of our simulation with regards to both geometry and radiometry in a series of controlled experiments. We first compare direct pixel-wise difference between images (as this is a direct measurement of the faithfulness of our data generator), and then study how this difference affects correspondence computation and reconstruction error for 3D scanning.

**Simulation Accuracy.**   For the first experiment, we computed a pixel-wise difference between the image acquired by the scanner of our *Pawn* calibration object and the simulated image. Figure 1 shows that the images are matching closely, with differences rarely exceeding $5\%$. Bigger errors of up to $15\%$ towards the bottom of the image were expected because of global light scattering of the

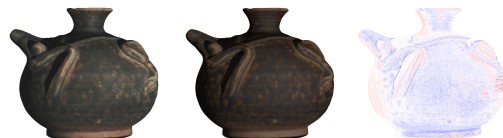

Figure 3: The image of the colored object (left) is very similar to the simulated image (middle). The colors are different due to the lack of calibration during 3D printing. We show the normalized pixel-wise differences in luminance on the right, with the same color scale as used in Figure 1.

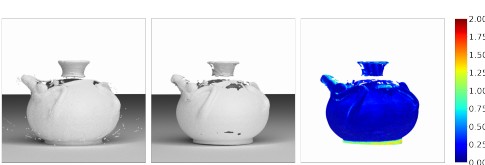

Figure 4: Point clouds from real and simulated scans and their error. The mean error measured with one-sided point-to-point distance is $0.17\,\text{mm}$. Note that some parts of the reconstructed object differ due to lighter and less saturated colors of the 3D-printed object.

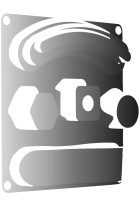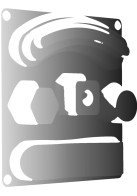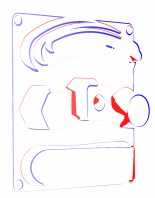

Figure 5: Color-coded (gray-scale) horizontal correspondence indices retrieved from the physical setup (left) and the simulation (middle) for Gray code scanning. The difference between the indices, linearly scaled between -1400 (blue) and 1400 (red) shows large errors only in boundary regions. The average index error is 0.97 (without boundary), and 12.3 (with boundary errors).

Figure 6: Distances between the reference geometry and the reconstructed point clouds from the physical scanner (left), simulated scanner before (middle) and after (right) optimization of material and projector parameters. We use the same colormap as in Figure 4. The larger noise level in the right image is due to less samples during the rendering.

rotating stage which was not included in the simulation. The difference is more visible when we switch to a 3D printed object having a different material and a larger fabrication tolerance (Figure 3). After reconstruction of the point clouds however, the geometry of the scanned and simulated version match closely – with a median error of $0.1\,\text{mm}$ and a mean error of $0.17\,\text{mm}$ from the simulated scan to the real scan (Figure 4).

**Correspondence Accuracy.** As an indirect validation, we compute correspondences from a full stack of coded images for both simulated and real images, and show the pixel-wise difference in Figure 5. Similar to before, the differences are rather small with correspondence indices being 0.97 off on average when regions where only one of the index maps stores an index are excluded. The boundary indices are less precise due to different causes such as misalignment, CNC-chamfering, and difficult thresholding on inclined regions.

**Reconstruction Accuracy.** Finally, we investigate the reconstruction errors after triangulation and use the CAD model from which the object was fabricated as the ground truth. For this, we calculated the point-to-mesh distance between the reconstructed point clouds and the properly positioned reference geometry. The results can be seen in Figure 6. Similarly to before, the results show that the simulation reproduces (most of) the artifacts and errors introduced by the physical scanner with respect to ground truth data.

## 5 Data Acquisition and Synthetic Dataset Generation

**Data Acquisition.** We scanned a collection of objects using our hardware setup: 4 calibration objects and 7 colored 3D printed objects. For each object, we acquired 12 views, rotating the object around its up-axis in steps of 30 degrees, at a resolution of 6464x4852. A typical acquisition time per object is several hours depending on the exposures used for HDR image capture. For all objects we also provide the reference geometry, allowing to use this data and mix it with synthetic data generated

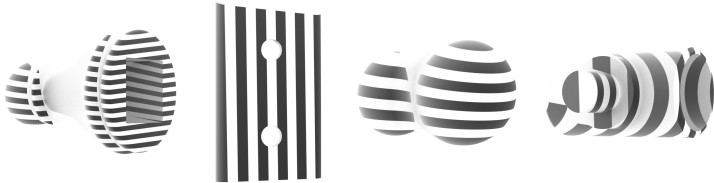

Figure 7: Four example CAD models illuminated with Gray code patterns from our synthetic dataset.

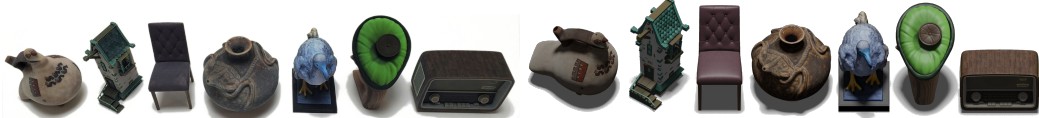

Figure 8: The 3D-printed textured objects (left) and the respective simulated renderings (right).

using our simulator. With this collection of acquired data, it will be possible to benchmark algorithms without having to reproduce the hardware setup. For further details about the acquired scan data, see the full description in the supplementary material.

**Software and Hardware Specifications.** The calibrated simulator is distributed as source code and a Docker container. It supports the use of custom light coding patterns, modification of scanner geometry (e.g. the baseline between camera and projector), editing of camera and/or projector resolution, and scanning of arbitrary objects provided as densely triangulated surfaces. It is divided into three parts: (1) The rendering pipeline that reads in geometry, projection patterns, material and transformation type (fixed pose for matching, turntable rotation for data generation and scanning, and random rotation for calibration purposes and data generation). (2) The decoding pipeline that takes the rendered images and decodes the patterns with a decoding function (we supply decoding for Gray code, unstructured light [2], and micro-phase shifting patterns [6]). (3) A reconstruction pipeline that reads in the rendered images from step 1 and the correspondences from step 2 and produces the reconstructed point cloud (with normals and colors) and depth map as well as the ground truth triangle mesh and depth map. In addition, we provide the hardware blueprints and the code for rerunning the scanner calibration, to allow the complete reproduction of our system and results.

**Large Scan Dataset.** Our dataset contains the physical scans of 4 calibration objects and 7 textured 3D printed objects (see Figure 8). We enrich this dataset with simulated scans from 1000 mechanical objects from the ABC dataset [11] (see Figure 7) with possible future extensions becoming available on our website. Each object is scanned with Gray code patterns in 10 random rotations in simulation and from 12 different directions on the rotating stage for the physical setup. In total, the dataset contains 192 physical (some calibration objects are scanned in multiple configurations, e.g. with or without ambient illumination) and 10k synthetic scans, annotated with ground truth depth and 3D geometry. For convenience, the synthetic dataset is split into multiple chunks of 250 scans each. For one chunk, the processing time on a 6-core machine was roughly 50 hours and is mostly depending on the render-resolution and the amount of samples per pixel. To the best of our knowledge, this dataset is unique and offers a next level benchmark for the evaluation of traditional and data-driven algorithms for processing 3D geometry (Section 6). We also envision this dataset and the scan simulator to become useful for developing postprocessing, correspondence, and texture reconstruction algorithms. Our data does not contain personally identifiable information or offensive content.

## 6   Benchmarks

The scanning datasets generated using our simulator contain ground truth and, at the same time, are faithfully reproducing the errors introduced by a real scanner, such as camera/projector distortion and defocus, radiometric errors, and decoding errors. We introduce a set of three benchmarks, two targeting low-level processing of individual range scans (which due to their regularity are a good match for CNN-based approaches) and the last one targeting surface reconstruction, a problem requiring unstructured approaches to scale to high resolution.

Table 1: The trained denoising model reduces the mean absolute error for synthetic and physical test scans and outperforms the non data-driven methods. Since the loss function doesn't consider normal orientations, the traditional methods outperform CNN with respect to this metric.

| | ↓ MAE [mm] | ↓ RMSE [mm] | Normal Angle Difference | | | |
| | | | ↓ Mean [°] | ↑ 3 [%] | ↑ 5 [%] | ↑ 10 [%] |
|---|---|---|---|---|---|---|
| Scan | 0.6906 | 6.6553 | 26.20 | 2.73 | 6.48 | 19.85 |
| CNN | **0.6199** | 6.6111 | 19.69 | 5.92 | 14.16 | 38.29 |
| Bilateral | 0.6851 | 6.6374 | **3.79** | **77.69** | **88.53** | **95.29** |
| Laplacian | 0.6874 | **6.6072** | 6.72 | 45.86 | 67.40 | 86.94 |

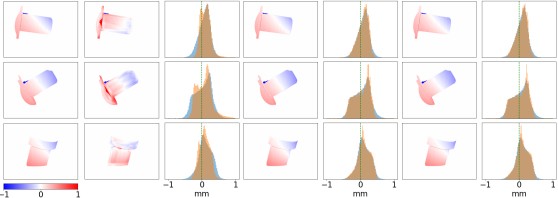

Figure 9: Errors of virtual scans before and after denoising. From left to right: signed error of the depth map recovered by SLS, signed error and distribution after depth denoising with our machine learning model, a bilateral filter, and Laplacian smoothing.

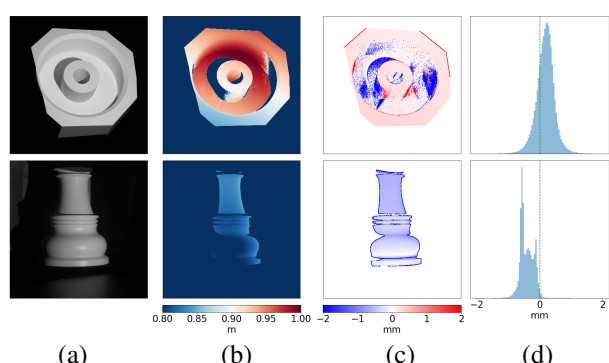

Figure 10: Results of our denoising model on a depth map from the physical scanner. The model was trained on synthetic data. From left to right: Depth errors after reconstruction, after denoising, improved regions.

## 6.1 Data-Driven Processing of Depth Images

The data we generated, containing real-world scan and highly faithful simulated data (both with ground truth geometry), is ideal for benchmarking low-level processing of depth scans. We propose benchmarks for two common tasks: 1) scan denoising to remove artifacts that arise from calibration problems and noise in the scanning process and 2) hole filling to complete gaps in the depth images. We also provide the results of baseline methods for each task. The latest version of the benchmark datasets, baseline results, and further details about the benchmarks are available on our website (and in the supplementary material).

**Denoising.** Minor inaccuracies in the calibration and errors in triangulation result in subtle errors in the reconstructed depth (see Figures 11 and 12). In contrast to outliers that can occur when scanning highly specular objects, these imperfections are located on a much smaller scale (millimeter to submillimeter), a challenging problem that is very relevant for metrology applications. Despite the large body of work in machine learning and depth map processing, we are unaware of any related work in the structured-light scanning context that tries to correct errors at this scale in a data-driven way, likely due to the lack of a data generator or acquisition setup for this problem. We use our approach to build such a dataset and propose a baseline method that casts the task as an image-to-image translation problem and train a convolutional neural network on pairs of reconstructed and ground truth depth maps.

(a)      (b)      (c)      (d)

Figure 11: Small-scale depth error in the scans of a virtual sample from our synthetic dataset (top row) and the real *Rook* calibration object (bottom row). Camera image with white projector light (a), the reconstructed depth map (b), its signed error to the ground truth depth (c), and the error distribution (d). Errors larger than 2 mm are clamped in the error maps and are not considered in the distributions.

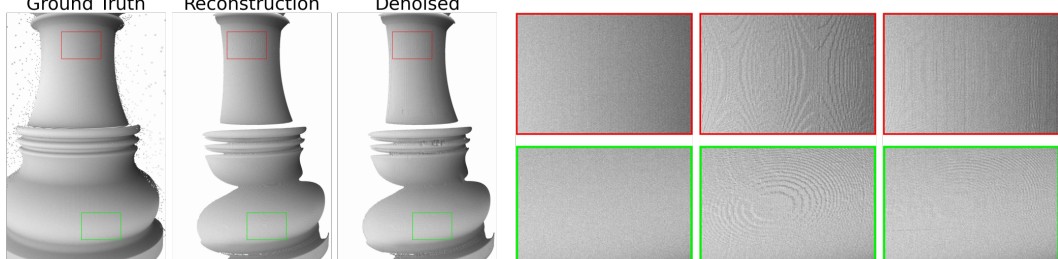

Figure 12: Point clouds of real scans before and after denoising with our network trained purely on synthetic data. The staircase artifacts in the SL scan are significantly reduced.

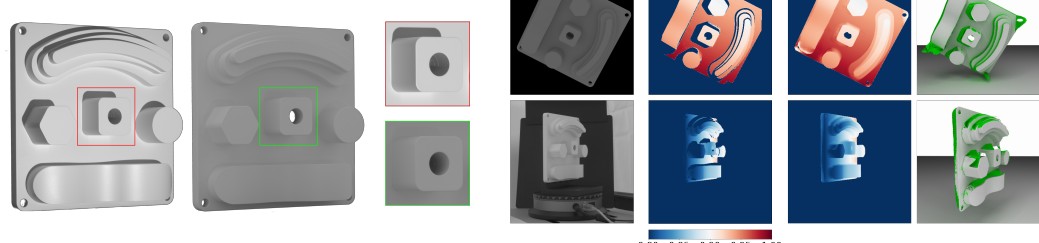

Figure 13: From left to right: the *Shapes* object illuminated with white projector light and ambient light (which can aid the reconstruction tasks). Both images are acquired with the hardware scanner.

Figure 14: Shape completion results for a synthetic (top) and a real scan (bottom). From left to right: Ambient image, reconstructed depth map, reconstructed depth map with filled holes, and corresponding rendering with green indicating filled areas.

The example training dataset is composed of 250 samples of reconstructed and ground truth depth maps. We tested three baselines for this task: a simple bilateral filter whose parameters are tuned with a sweep on the training data, a mesh-based Laplacian smoothing that is constrained to move the depth map vertices parallel to the camera's optical axis, and a novel data-driven approach inspired by other depth denoising [22] and image-to-image translation approaches [7]. The details on the latter approach are in the supplementary material.

Table 1 and Figure 9 show the benefit of the data-driven approach compared to the two baseline methods. The generalization of the model trained on synthetic images to images from the physical scanner is shown in Figure 10. As for the synthetic test objects, the model improves the error distribution by reducing its standard deviation. Note that we can compute the error of the real *Rook* scan because for the calibration objects we have established an accurate alignment to the simulated environment. The point clouds in Figure 12 show the reprojected depth maps in 3D space. Typical reconstruction artifacts like the staircase pattern are removed or partially attenuated in the denoised scan, which confirms that the model learned to correct specific artifacts of the real scanning process by analyzing statistics of the virtual data.

**Shape Completion.** Shape Completion (sometimes also referred to as shape inpainting, hole filling, or depth completion) is a common postprocessing task in 3D scanning. In our baseline example, we generate 100k crop images from our dataset. Each sample consists of an image crop from the diffusely illuminated rendering, the corresponding ground truth depth map as well as the reconstructed depth map from the virtual scanning process. In contrast to images with projector illumination, the diffusely illuminated ones provide the network with information about areas not hit by projector light rays. See Figure 13 for examples of ambient and directed illumination and their difference. We used the data-driven approach by [21] for depth completion of indoor scenes as a baseline, see the additional material for more information on the training regime and architecture. For the training of their model, we modify the data loader to read samples from our synthetic dataset in the form of cropped patches of size $320 \times 256$. The training is run for 100 epochs. We did not further modify their approach, so we refer to the original paper for more details on the model and training process [21].

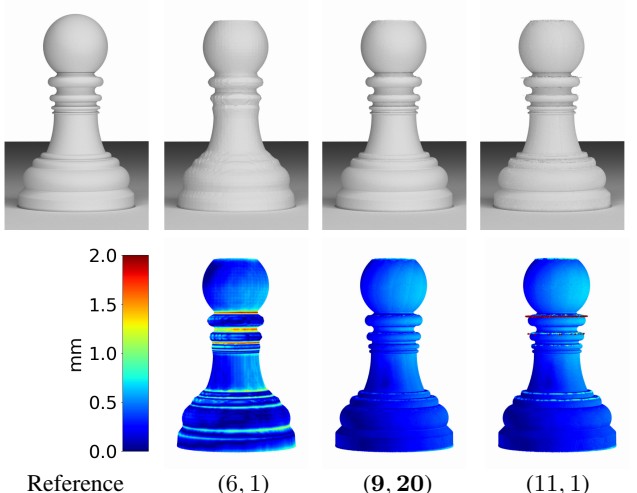

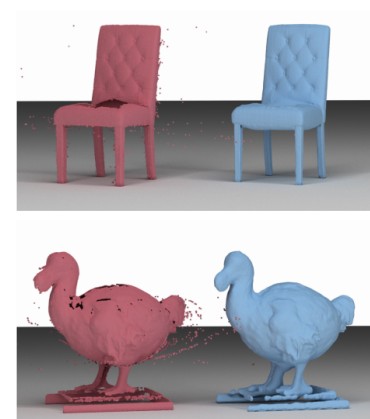

|  |  |  |  |
|---|---|---|---|
| Reference | $(6, 1)$ | $(\mathbf{9}, \mathbf{20})$ | $(11, 1)$ |

Figure 15: Error of reconstructed surfaces for selected parameter combinations. Reference mesh next to surfaces reconstructed from virtual scans (top row) and their error with respect to the reference (bottom row). The parameters $(D, N_S)$ with the lowest error are highlighted.

Figure 16: Point clouds (red) next to reconstructed surfaces (blue) for two test objects. The point clouds are from real scans and the surface reconstruction parameters determined from synthetic scans.

For inference on full resolution input, we apply the trained model on overlapping tiles and merge the results. The final completed depth map is a hybrid of the reconstructed depth map from the scanning process and the output of the trained model: We preserve valid depth values from the scan and use the model to fill in only the missing areas. To quantify which areas to fill specifically, we use an object mask extracted from the ambient image. Figure 14 shows shape completion results for both virtual scans from the simulated scanner and a real scan from the hardware scanner. The preliminary results suggest that the model trained purely on synthetic data can generalize to our real world 3D scans. The resulting depth maps have less holes from occlusion and are better suitable, for example, for surface reconstruction tasks. While these results still rely on an algorithm introduced for indoor depth processing, our datasets open the door to future methods specifically designed for shape completion in the structured-light scanning context.

## 6.2  3D Reconstruction

A common downstream task in structured-light scanning is assembling a full model of a captured object. For this task, multiple depth maps are fused into one point cloud and a consistent surface is extracted from this unstructured representation. Our system can be used to simulate the scan of any 3D object. We provide scripts to automatically generate an arbitrary number of range scans and fuse them into a single point cloud that can be processed with a 3D reconstruction algorithm. The reconstructed geometry can then be compared with the reference geometry using Hausdorff distance to quantitatively measure the reconstruction error. We provide the script to generate the data and evaluate the reconstructions, and as an example we use them to evaluate the screened Poisson surface reconstruction (SPSR) [10] algorithm. This is not a data-driven approach, but its parameters are context dependent and have to be adapted according to the input data. We use our system to automatically find optimal parameters for a given object.

Table 2: One-sided Hausdorff distance (in mm) for different parameterizations of the Poisson surface reconstruction. $D$ is maximum octree depth and $N_S$ is the minimum number of samples per node.

|  |  | $N_S$ | | | | |
|---|---|---|---|---|---|---|
|  |  | 1 | 5 | 10 | 15 | 20 |
|  | 5 | 3.600 | 3.607 | 3.610 | 3.611 | 3.601 |
|  | 6 | 18.937 | 2.334 | 2.333 | 2.334 | 2.324 |
|  | 7 | 18.222 | 13.888 | 1.576 | 1.570 | 1.572 |
| $D$ | 8 | 16.531 | 13.667 | 1.365 | 1.394 | 1.380 |
|  | 9 | 17.871 | 13.869 | 1.334 | 1.326 | **1.308** |
|  | 10 | 18.390 | 15.828 | 1.337 | 1.342 | 1.354 |
|  | 11 | 18.416 | 16.187 | 1.345 | 1.348 | 1.328 |

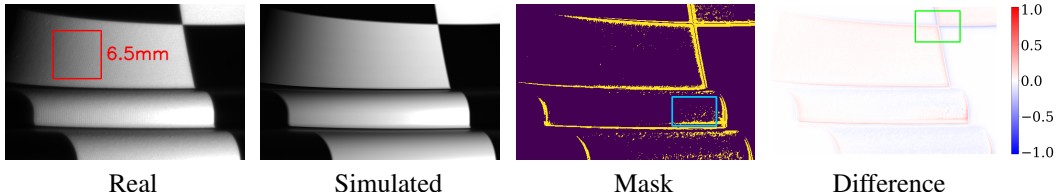

| Real | Simulated | Mask | Difference |

Figure 17: From left to right: close-up view of the highlighted red region of Figure 1 with corresponding rendering/simulation in non-inverted colors. The error threshold for the mask image (5%) and the scale of difference images are the same as on Figure 1. The remaining error sources include gaps between projector pixels ($\approx 2\%$, highlighted red), minor misalignment and difference in projector defocus around sharp transitions (up to 20%, highlighted green), and material surface microstructure in specular regions (about $5 - 10\%$, highlighted cyan). Relative sensor noise in the camera image is negligible ($< 3\%$ for dark regions) because of HDR image capture and, therefore, is not simulated.

We perform a parameter sweep for the maximum depth $D$ of the octree over which the equation is solved and the minimum number of samples $N_S$ in each octree node over the *Pawn* and *Rook* calibration objects. Table 2 shows the one-sided Hausdorff distances in millimeters from the meshes recovered with SPSR to the ground truth surface. The values are averaged over the runs for both test objects. Figure 15 provides qualitative insights for some parameter combinations and the error distribution over the surface. Note that we trimmed the reconstructed meshes above and below a certain height to avoid limitations imposed by the capture setup (e.g. we cannot reconstruct the bottom of the object as it is invisible). The parameters that achieve the best score in the sweep produce a surface that is very detailed, yet is not dominated by the noise in the input point cloud. We show that the parameters discovered in the virtual environment generalize to the real world by applying them to a set of point clouds acquired by the real scanner (Figure 16).

## 7 Concluding Remarks

We built an accurate structured-light scanner and a corresponding simulation pipeline that, after calibration, can reproduce pixel-accurate replica of the images acquired by the scanner's camera. The acquired data and the simulator allow us to construct 3D scanning datasets with ground truth, which are ideal for development of data-driven algorithms for surface reconstruction, providing a quantitative criteria to evaluate their performance. Additionally, our simulation pipeline can be used for structured-light scanning algorithm and coding pattern development. Our results also demonstrate the lower bound of achievable simulation accuracy (Figure 17) even in a very simple setup, without resorting to extreme measures like simulating the internal structure of the light sources or accounting for thermal expansion. However, we did have to simulate the effects of diffraction limit and chromatic aberrations to achieve the demonstrated level of matching, which illustrates the importance of validating the simulator against a real-world physical scanner. We do not foresee any potential negative societal impact of our work.

**Limitations.** The high accuracy of our system has been achieved by controlling the lighting setup and restricting the choice of materials. Despite its apparent simplicity, we believe this is an important setting, as it allows to establish a baseline performance for 3D reconstruction methods tailored to structured-light scanning. For instance, we were able to estimate, quantitatively, the effect of global illumination due to the secondary light scattering of the rotating stage supporting the scanned object (Figure 1). See our supplementary material for a more detailed explanation of error sources and limitations of the setup.

**Improvements.** A potential improvement to our simulator is inclusion of the varying blur kernels for projector defocus simulation. Projector focus and aperture calibration stage revealed that the defocus pattern for projector pixels changes across the projected image and with distance away from the focal plane. It is the largest remaining source of errors in our simulator (Figure 17). However, accurate simulation of this effect would complicate the calibration procedure significantly. Extending our work to more realistic lighting setups and to materials with non negligible subsurface scattering is an exciting avenue for future work too.

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
