# OpenReview forum: "Hardware Design and Accurate Simulation of Structured-Light Scanning for Benchmarking of 3D Reconstruction Algorithms"
_NeurIPS.cc/2021/Track/Datasets_and_Benchmarks/Round2 — NeurIPS 2021 Datasets and Benchmarks Track (Round 2)_

### Official Review · Reviewer_qd7E · 2021-09-13
**Exciting direction of work, while benchmarks make assessing its benefit difficult.**

**Rating:** 6
**Confidence:** 3

**Strengths:**

The broad theme of optimizing calibration and material in synthetic environments to improve rendering is exciting to the field of 3D vision, as fully-supervised 3D is challenging to obtain on realistic-looking images. This paper provides a small step towards this end.

This setup enables the unique situation of data-driven scan denoising, and qualitative results are impressive. Similarly, data-driven shape completion enables results such as highly detailed hole-fillings in Figure 14.

Hardware setup and data gathering are meticulous. They yield a large set of HDR images for which precise 3D data is available.


**Weaknesses:**

Limitations in scope: while I acknowledge collecting data at such precision is difficult and scaling up may not be feasible, the following two points are nevertheless limitations of the paper.
1. Minimal real data: this dataset consists of only 10 real objects. I acknowledge the scans have high detail and therefore memory, but from a practical standpoint 10 objects is a very small sample to draw any “real-world” data conclusions.
2. Highly-controlled, limited data setting: the paper refers several times to scenes – in 3D this is where we most need realistic and annotated data. However, in addition to real objects being limited in number, they are indeed single objects, not scenes, are of approximately uniform size, and have existing CAD data available. This limits the method from being applicable beyond a small domain.

Evaluation of setups has no baselines or ablations: although reconstructions look good, it is difficult to assess the performance without comparisons to prior work. More explanation is needed. See Correctness.

The paper explains the proposed data-driven method underperforms simple baselines due to lack of normal orientations. However, this does not give evidence in favor of data-driven scan denoising.


**Additional Feedback:**

None at this time.

**Clarity:**

In general, the paper is well written and easy to follow. Please see my main criticism of comparison to related work below, which confused me at first read.

**Correctness:**

While the data collection is well-described, its benchmarks are not. Instead of presenting an experiment to compare different methods on an important task, the benchmarks typically run a out-of-the-box method and show one or two qualitative results. This seems unlikely to be broadly useful to the scientific community, and does not help back the paper’s claims of facilitating data-driven training on high-resolution 3D.

For instance, line 268-271: “The preliminary results suggest that the model trained purely on synthetic data can generalize to our real world 3D scans. The resulting depth maps have less holes from occlusion and are better suitable, for example, for surface reconstruction tasks.” Yet, these results refer to a single figure, which is a qualitative example and contains no comparison or ablation.


**Documentation:**

All answers appear to be yes. The authors detail data collection and share URL to access the dataset.

**Ethics:**

I do not have ethical concerns.

**Relation To Prior Work:**

Fair. The paper overviews relevant work across a few subfields, which is helpful as the paper is somewhat cross-disciplinary. However, limited reference to prior methods used during evaluation e.g. Image-To-Image Translation or Shape Completion. One additional drawback is that I am unable to find citation or detail for “ABC” dataset contained in the paper. This is a critical component of the paper; please include in final version.

**Summary And Contributions:**

This paper proposes a technique to jointly scan an object and render realistic images. A dataset is gathered using the technique consisting of 3 calibration objects and 7 3D printed objects, of which CAD models are available. Scans are also simulated for 1000 objects from the ABC dataset. Careful setup along with optimization of calibration parameters and materials enables high rendering accuracy of the simulated environment, enabling supervised 3D models to train in on realistic images.

---

> ### Author Response · Authors · 2021-09-26
> **Answer to your feedback**
>
> We believe that there is a misunderstanding in the goals and scope of our contribution that we would like to clarify. Our main contribution is a data generator, the dataset itself is only supposed to be used for calibration of the simulator and to test generalization to real data.
>
> **Minimal Real Data.** The main advantage of our approach is that it allows the creation of synthetic data that is indistinguishable from acquired data. While we agree that having more than 10 models would be useful as additional testing data, we do not think this will make much of a difference as the purpose of the acquired data is to calibrate the system and, only after a data-driven model is trained, to check for generalization to real data.
>
> **Highly-Controlled, Limited Data Setting.** Our simulator does support arbitrarily complex object geometries: geometric complexity is not an issue and it can be used on scenes composed of many objects. As we point out in the limitations, the selection of lighting setup and material for which we tested pixel-wise reproduction is limited. However, our system can be used to generate synthetic data for other more complex setups, but in that case, we cannot ensure pixel-wise accuracy. We are planning to slowly extend the selection of materials, but this is an ongoing work that we argue is beyond the scope of this paper: it took around 2 years of work to design the hardware and software to obtain pixel-wise accuracy on our current scenes as it is a highly non-trivial problem, despite the apparent simplicity of the scenes.
>
> **Baselines and Ablations.**  An extensive benchmark for each one of our applications would be definitely a valuable addition to our work. The purpose of our baseline comparison is not to establish which method is SOTA for any of our problems, but rather introduce a dataset and challenge so that other researchers can test their algorithm. Our goal is to demonstrate that our data generator can be used to do quantitative analysis of these problems, and used to generate data to train supervised machine learning methods: we thus argue that one baseline for each application is sufficient. We plan to extend our benchmarks over time, but this is out of scope for the current work.
> We are confident of the conclusions we have for the baselines we considered (and clearly they do not extend to all SOTA methods in these areas), but we could downplay further (or remove) our conclusions on all three applications if the reviewers believe that our analysis is too preliminary to be published.
>
> **ABC Dataset.** We apologize for the omission, the citation has been removed as we were shortening the paper to fit in the space limitations. We will add it back: https://deep-geometry.github.io/abc-dataset/

---

> > ### Comment · Reviewer_qd7E · 2021-09-29
> > **Final Review**
> >
> > Hello. Thanks to the authors for the rebuttal. After discussion with the reviewers, I have decided to increase my rating to 6 - marginally above acceptance. Like some other reviewers, I have some concern about the comparisons to prior work and believe the title should be changed. However, from the rebuttal I give more weight to the proposed generation pipeline and thus change my rating towards accept.
> >
> > Regards,
> > R3

---

### Official Review · Reviewer_Pbmn · 2021-09-19
**Review for ID 75**

**Rating:** 6
**Confidence:** 4
**Correctness:** The proposed dataset is constructed i…
**Clarity:** The paper is clearly written.

**Strengths:**

+ The 3D dataset (10 objects) that includes both scanned results and CAD models are useful to quantify the performance of existing 3D reconstruction and postprocessing algorithms.

+ A novel system and associated calibration script can be used to simulate the scan of any 3D object. This allows a user to customize their dataset with their own 3D models.

**Weaknesses:**

- Weak justification for the novelty of the proposed dataset. The novelty of this data comes from the fact that it provides a 3D CAD model and the paired real scan. However, it is not fully justified why we indeed need re-scan of the 3D CAD model. Why not simply reconstruct a 3D geometry from a set of synthesized 2D images (e.g., multiview stereo from synthesized multiview images) from a 3D CAD model?

- Weak scalability in data capturing pipeline and lack of dataset. The data capturing pipeline is not practical to collect a large amount of data. It requires printing out a physical object and re-scanning the 3D-printed objects. This leads to collecting a small amount of textured dataset (ten objects). Compared to the previous 3D model dataset [1,2,3,4,5], this paper shows weakness in the size and diversity of the dataset.

[1] 3D human body dataset: https://dfaust.is.tue.mpg.de/,
[2] Non rigid object dataset: http://tosca.cs.technion.ac.il/book/shrec.html
[3] 3D object dataset: http://modelnet.cs.princeton.edu/
[4] 3D chair and vehicle dataset: https://shapenet.org/
[5] Many textured 3D cad model dataset: https://github.com/timzhang642/3D-Machine-Learning

- Insufficient discussion on the previous 3D dataset and weak comparison with them. The current related work and experiment are not sufficient to localize this paper to the state of the art. A deeper discussion and analysis on why existing datasets are not sufficient, what their limitations are, and how this paper addressed those limitations are required to highlight the novelty of the proposed dataset. For example, 1) the number of dataset and their types can be summarized in a table; 2) a paragraph that discuss why the proposed dataset is better than others in terms of what perspective is needed; 3) a cross dataset evaluation can be made using multiple 3D model datasets (e.g., training a model on the previous dataset, and testing on the proposed dataset).

- Weak formulation on benchmark challenges. The suggested three benchmark challenges do not effectively demonstrate the quality and novelty of the proposed dataset. Most importantly, it seems possible to enable the suggested challenges even without the proposed data. The current benchmark formulation does not answer the following questions. 1) Denoising tasks: why not simply add synthetic noise on existing CAD models? More importantly, how meaningful is it to handle submillimeter-scale noise (i.e., the quality of the reconstructed surface already looks good)? 2) Shape completion task: why not simply add synthetic holes on the CAD model?; why can the previous 3D model dataset not be used for this task? 3) 3D reconstruction tasks: it is not clear what the authors wanted to deliver from this task. Why is it challenging to reconstruct 3D models from clean depth maps and calibrations?; why can the previous 3D model dataset not be used for this task?



**Additional Feedback:**

Please see the weakness section for the main comments.

**Documentation:**

This paper includes sufficient detail on the dataset collection pipeline.

**Relation To Prior Work:**

It is clear how the system differs from previous works, but not clear the dataset. It would be nice if the authors provide a clear summary of the number of dataset and its category in the context of previous works.

**Summary And Contributions:**

This paper introduces a novel hardware system along with a rendering pipeline to collect a benchmark dataset for 3D reconstruction and preprocessing tasks. Overall, three calibration objects and seven textured scanned objects are provided. Three benchmark challenges are designed to advance data-driven models for 3D reconstruction and geometry processing: 1) surface reconstruction from point clouds. 2) denoised 3D geometry reconstruction from noisy geometry. 3) 3D shape completion from partial geometry. For each task, several existing methods are evaluated as initial baselines.

---

> ### Author Response · Authors · 2021-09-26
> **Answer to your feedback**
>
> We believe that there is a misunderstanding in our goals and contribution that we would like to clarify.
>
> **Contribution.** Our major contribution is a data generator, which is sufficiently accurate to generate training data that is indistinguishable from real data acquired by our hardware system. The way we envision the use of our contribution, and the way our benchmarks are designed, is to use the generator to create the training data and then to test it on the real data  (consisting of a small number of representative samples) obtained using a scanning setup similar to ours. It is not expected that the scanning system will be used to scans thousand of models, our goal is to exactly avoid having to scan such a large number of objects by using a simulator, while at the same time providing synthetic data that is indistinguishable from the real one, thus capturing real-world noise and other imperfections and a way to calibrate the simulator and verify its consistency with real data by providing a scanner design.
>
> **Data Type and Mesh Datasets.** Our dataset is composed of a set of range scans with ground truth reference to the scanned geometry, and it is thus unrelated to datasets of 3D models such as Shapenet. We could add a discussion of the dataset papers mentioned by the reviewer, but we emphasize that they are not related to our work: a possible connection is that they could be used by our data generator instead of the ABC dataset as inputs.
>
> **Challenge 1: Denoising.** We strongly disagree that adding synthetic noise on CAD models yields adequate synthetic data for most applications. While there are many ML papers that are tackling this problem as a way to test the performance of ML algorithms, we don’t believe that this is an approach adequate for practical applications. In our experience, ML methods trained on synthetic data (with artificial noise added) generally do not perform adequately on real scanned data, due to different noise distribution. One can say that developing a model of noise that matches closely the actual noise for the physical scanner we’ve developed as a part of the project is one of the important goals of our work.
>
> **How meaningful is it to handle sub-millimeter scale noise (i.e., the quality of the reconstructed surface already looks good)?** We believe it is very important, as the assessment of quality is highly application-dependent. Even in the context of  graphics applications for which “looking good” is the defining consideration, the quality standard varies in a broad range depending on quality.  Our work strives to make quantitative evaluations possible enabling comparison of different reconstruction methods.
> We also note that the accuracy requirement is also best defined relative to the scene size that is being scanned instead of in absolute units. For instance, when scanning an object 10 cm in height (as in our setup), 1 mm constitutes an error of 1%, so sub-millimeter accuracy is a must for high fidelity reconstruction. However, when scanning a room on a scale of 10 m (e.g. with LIDAR sensor), sub-millimeter accuracy may be an overkill (less than 0.01% of the scene size).
>
> **Challenge 2: Hole Filling.** Similar to challenge 1, we argue that adding artificial holes to a 3D model is not representative of missing areas in real 3D scans. It is a simple way to generate synthetic data to explore ML methods in tasks where capturing real data is difficult and expensive. Our approach produces data indistinguishable from the data acquired from a real scanner, thus any missing areas in synthetic scans are similar to real ones.
>
> **Challenge 3: Surface Reconstruction.** We would like to point out that reconstruction of surfaces from high-resolution range scans with low noise is a widely used technology in computer vision and computer graphics with applications in manufacturing, special effects, cultural heritage preservation and many other areas (we estimate that it is the most commonly used approach). Reconstruction in this setting still suffers from a broad range of problems, and is a topic of active research.  An example of a recent paper in this area is https://arxiv.org/pdf/2006.13782.pdf and a survey is available at https://matthewberger.github.io/papers/reconstar.pdf. Our work is the first method that allows objective, quantitative evaluation of reconstruction errors and will thus be an important component to assist researchers in this area.

---

> > ### Comment · Reviewer_Pbmn · 2021-09-29
> > **Reply to the authors**
> >
> > Dear authors,
> >
> > thanks for the detailed response to the review. After reading the rebuttals, many of my concerns are clarified. In particular, I accept that I should put more weights on the proposed dataset generation pipeline which will be provided to a user to create their own synthetic dataset, where this generators can ensure the quality of real scanned dataset. Still, however, I believe the experiment and benchmark challenges are weakly designed and the number of provided dataset is lacking to formulate a benchmark challenge.
> >
> > By weighting the down/upsides of this paper, I would like to raise the rating.
> >
> > Upon acceptance, however, the authors should provide a clear and well-structured guidance to a user about how to use the proposed system to generate the synthetic dataset as highlighted by this paper.
> >
> > Regards,

---

### Official Review · Reviewer_9jeD · 2021-09-21
**solid work on structured light scanning hardware and simulator co-design for data capture and generation, but also restrictive to this scanning setting**

**Rating:** 6
**Confidence:** 3
**Clarity:** well written

**Strengths:**

- a solid setting for benchmark 3d recon for structure light scanning, including both hardware and simulator, and plenty of details on the hardware, renderer, etc in the supp materials
- rigorous data gathering pipeline, and detailed data generation process, presented in the supp material, including preprocessing, data source, code to reproduce
- really like the docker image and try out the jupyter notebook for rendering scan images, decoding and reconstruction


**Weaknesses:**

- the title is a bit overstating the scope of this paper, it seems the current capture, rendering and datasets are all for structured light scanning, but not 'general' 3d reconstruction, also the abstract does not explicitly point it out, but tell the story from a very abstract level
- as pointed out in the supp material, the 3 benchmark problems using this dataset/setup are all postprocessing algorithms/models for 3D scanning, which are decoupled from the specific scanning mechanism, e.g. ToF, LIDAR, etc. But it is not clear how useful the datasets generated from SLS can be transfer to other scanners? e.g. what if I have a room level scanning data from iPad pro, point cloud of a outdoor driving scene?
- this is stated in the limitation section, there is not much variation of difficulty or diversity of lighting and material


**Additional Feedback:**

- L49: difference between 3 precisely-machined calibration objects and 7 color-textured 3D printed objects? any considerations for choosing those particular objects as the testset? are they diverse enough as testset to evaluate the generalization of new reconstruction models?

- L103: "We are not aware of any existing work able to achieve a faithfulness comparable to our approach, especially on geometrically complex objects"  --- this is a bold claim

- L250: what's the resolution of 250 samples? bitdepth? Is it too small as training set? what the distribution it can represent? could it too restrictive to generalize only to the CAD models from ABC dataset?

- Table 1: CNN performs too bad on normal direction, have you consider surface normal loss additional to the depth loss?

- Fig 11: it looks like denoised depth introduces large positive error (red points). is it expected?

- L254: can I understand the Shape completion task considered here is grayscale image (diffusely illuminated rendering) guided depth completion?

- L269 & Fig 14: the generalization is shown from synthetic to real on CAD models, how about to the real objects?

- I run the ipynb for simulating scans, decoding and reconstruction. It’s a very nice and smooth experience. Any idea for why the recon for vase misses point clouds near the neck?

**Correctness:**

I am not familiar with SLS, can not comment on the correctness of hardware setup and rendering (w/ mitsuba)

**Documentation:**

datasets, code are all openly accessible, license is clear

**Ethics:**

no problem

**Relation To Prior Work:**

I think there are lots of sim-to-real works in robotics / self-driving, so the claim in L94 is a bit bold

**Summary And Contributions:**

This work propose to adapt the simulator parameters to one specifically design structured light scanner so that the rendered scans are close to the captured scan images. The tuned simulator can be used to generate 10K photorealistic scanning images for 1000 CAD objects from ABC dataset under this hardware setup, which can be used to train and benchmark post-processing models/algorithms for 3D scanning, particularly depth denoising, depth completion and surface reconstruction from point cloud. Besides those synthetic data, this work also provides real captured scans for 4 calibration objects and 7 color textured objects (3D printed from models on Sketchfab).

---

> ### Author Response · Authors · 2021-09-26
> **Answer to your feedback**
>
> **Title change.** We will change the title to “Hardware Design and Accurate Simulation for Benchmarking of 3D SLS Reconstruction Algorithms” to make for a clearer contribution.
>
> **Specialized for SLS.** We will clearly state in the introduction and in the conclusions that our focus is on SL scanning and reconstruction. We agree that extending our work to other scanner types would be useful for the community, but we consider this future work.  Practically achieving pixel-wise matching is a substantial engineering effort, including identifying experimentally all significant factors affecting the final reconstruction, and designing suitable simulation and calibration methods to account for these factors. The overall effort required around 2 years of effort with two PhD students dedicating a significant fraction of their time to the project. We expect that other acquisition modalities will present similar, possibly greater challenges. For instance, camera lens distortions can be corrected for in the images captured using physical setup whilst lens distortions of the projector are essential to simulation as they affect the way the light is being injected into the scene (attempts to correct for projector distortions by means of pre-distorting the projected pattern used in physical setup would result in severe loss of contrast due to the limited projector resolution when there is no such limitation in simulation).
>
> **I think there are lots of sim-to-real works in robotics / self-driving, so the claim in L94 is a bit bold.** We agree with the reviewer that there is indeed work in which simulated datasets are used to train models that are applied to real data, and we review some of these works in the paper. However, an ideal synthetic training set should be indistinguishable from real data.  All existing approaches, to the extent we know, did not  produce synthetic data indistinguishable from real (neither were aiming to) and can be seen as domain randomization strategies. The goal of our work was to explore the question of what it takes to generate such perfectly realistic data for 3D reconstruction problems. While this appears to us a natural direction of research, given the growing importance of synthetic datasets in machine learning, we were unable to identify any previous work focusing on this, for 3D reconstruction or related problems. We would be happy to include references to any works we have missed - we would greatly appreciate it if the reviewer could provide references. For the final version, we can further clarify our focus and include a related work section on sim2real approaches.
>
>
> **Additional Feedback.**
>
> **L49:** The three CNC-machined objects have precise geometry and material properties and were designed for the calibration between simulation and physical scanner. The seven 3D-printed objects were chosen to show the reconstruction of textured objects and were therefore an interesting test case. There are other interesting cases (translucent, transparent and reflective materials) that can be tested with our setup in future work.
>
> **L103:** We can tone down this claim and would be happy to include other related work that shows results of similar faithfulness.
>
> **L225:** The resolution of the training images is 3232x2426. The direct and diffuse illuminated  images are 8-bit grayscale, the reconstructed depth maps are 32-bit float. Using a larger set would certainly increase the model accuracy and generalization and we could update the results in the final version with such an experiment.
>
> **Table 1:** Yes, we have included surface normal loss in our current implementation and would be happy to update these results in the final version of the paper.
>
> **Fig 11:** The large positive errors are artifacts that come from depth discontinuities in the reconstructed depth map.
>
> **L254:** Yes, the inputs to the shape completion task are diffusely illuminated images together with the reconstructed (incomplete) depth map from the structured light scan. The ground truth depth map can be used for training.
>
> **L269:** The goal of this experiment was to show the shape completion for untextured objects with simple materials. We did not evaluate the model on the textured object scans.
>
> **Simulation Software:** For the vase scan, the parts at the neck of the vase are not reconstructed since they are occluded during the scanning process.

---

> > ### Comment · Reviewer_9jeD · 2021-09-30
> > **thanks for the rebuttal, I keep my rating as is**
> >
> > Thanks the authors for the rebuttal. Like other reviewers, I think the SLS hardware and simulator co-design is very solid work, but has the concern of the utility of the restrictive setup and generated data to broader community in 3d recon and understanding. Also the benchmark is weak. As for the sim2real literature, one early work I remember is this one, but in the distribution level, not instance level. (I am not related to this paper)
> > - Shrivastava, Ashish, et al. "Learning from simulated and unsupervised images through adversarial training. CoRR abs/1612.07828 (2016)." arXiv preprint arxiv:1612.07828 (2016).

---

### Decision · Program_Chairs · 2021-10-09

**Decision:**

Accept

**Comment:**

The paper develops a 3D scanning setup and rendering pipeline that allows for near pixel-perfect matches between photographs of the scene and renders of the scene. The paper argues that this enables researchers to develop models for various 3D tasks on simulated data generated from their pipeline, and introduces three benchmarks based on simulated data. All reviewers agreed that the task is important, and that the careful co-design of the scanning hardware and rendering pipeline are technically impressive. Reviewers were initially put-off by the small number of actual scans provided, but after discussion agreed that the emphasis and true value of the work are not the scans of actual objects, but instead the data generation pipeline that allows for generating highly realistic synthetic data. In the end all reviewers were marginally in favor of accepting the paper. The AC agrees with the conclusion reached by the reviewers, and thinks that the synthetic data generated from the proposed pipeline is interesting and likely to be useful. Congratulations on having your paper accepted to the NeurIPS 2021 Datasets & Benchmarks Track!

When preparing the camera-ready version of the paper, authors are strongly encouraged to take into consideration the feedback from reviewers. In particular, the AC agrees with reviewers that the title of the paper should be changed (as the authors have already agreed to do); the authors should also consider revising the paper to better emphasize the importance of the synthetic data generated by the pipeline.